# How to Make E-Commerce More Successful by Use of Kano's Model to Assess Customer Satisfaction in Terms of Sustainable Development

**Manuela Ingaldi \***  **and Robert Ulewicz**

Faculty of Management, Czestochowa University of Technology, al. Armii Krajowej 19b,
42-200 Czestochowa, Poland
* Correspondence: manuela.ingaldi@wz.pcz.pl or manuela@gazeta.pl; Tel.: +48-34-32-50-449

**Abstract:** Personalization, mobility, deliveries on the same day, and perhaps artificial intelligence—all of these elements will shape e-commerce in the near future. It is necessary to consider what features and standards online shops will have to meet in order to achieve success and to adapt to the changing preferences and requirements of the customer and their awareness of the perception of the environment through the prism of, for example, sustainable development. This means there is a need to specify a set of attributes that will influence the decision to use the services of a given e-shop. Despite all efforts, many online shops fail because they do not meet the expectations of customers. At the same time, meeting customer expectations is a big challenge for newly emerging e-shops. There are many studies on sustainable development in e-commerce, but there is no specific methodology for e-shop design, especially in the aspect of sustainable development. The authors propose a methodology based on Kano's model and customer satisfaction to explore customers' stated needs and unstated desires and to divided them into different groups with different impacts on customer satisfaction. In this paper, a case study on the attributes of customer satisfaction for a newly opened e-shop with organic products, which is to operate in the countries of Central Europe and takes into account selected assumptions of sustainable development, was presented. The research took the form of an original (authorial), universal survey that can be used in other similar research. A total of 1069 correctly completed surveys were taken into account for the analysis. Respondents indicated 16 must-have features for the e-shop in order to make customers benefit from its services and 11 one-dimensional features that will determine the level of customer satisfaction. Among the must-have features, there were those related to sustainable development, which indicates the environmental awareness of potential customers. The obtained results were given to the management of the research e-shop and were included during the design of its operation. After one to two years of e-shop operation, the results will be verified.

**Keywords:** e-commerce; sustainable business; social challenges; Kano's model; small and medium enterprises

---

## 1. Introduction

In recent years there have been major changes in many industries, which represent a concept known as Revolution 4.0. This revolution has an impact on the functioning of all enterprises on the market and also forces them to make internal changes. It is caused by rapid development of available technologies, digitalization, and a huge amount of data. One of the tools that has allowed the development of Revolution 4.0 is the Internet, without which many young people cannot imagine their lives. On the Internet it is possible to do many things, and it has become something important and necessary [1].

People make smaller or larger purchases every day. At the beginning of the Internet development era, the first e-shops were created. Their offers, operation, forms of payment, and interface were subject to constant changes, adapting to the constantly changing requirements of their customers and markets. It can be noticed that competition in the e-commerce market is growing, and the fight between e-shops is taking new forms. It is believed [2] that in the near future, customers will, more and more often, make purchases in this form, and this will also apply to ordinary food purchases. New e-shops will also appear. The highly dynamic development of e-commerce puts this sector among the fastest growing in the European economy. It can be concluded, therefore, that e-commerce is a great opportunity for this economy [3,4]. There is also a combination of different retail channels during the purchase process, as some customers tend to change sales channels when shopping for no specific reason [5–7]. All this is caused, above all, by omnipresent globalization and the digital revolution [8].

Another reason for such a large popularity of e-commerce is the omnipresent rush and lack of time. Customers prefer to sit close to the computer at home in the evening to choose products than to go to stationary shops, galleries that are full of people and products, and where sorting products is more difficult and time consuming. After all, things bought on the Internet can be checked at home, and in case of lack of satisfaction, they can be sent back to the e-shop. What is important is that many online shops offer free returns of sold products.

It is also important to emphasize the sociological aspect of the stimuli shaping future consumer attitudes. The market must adapt to changes and new customer requirements. For example, for the Z generation, Facebook is no longer an important part of life, as for Millennials. They prefer to use Snapchat or Instagram. Young people prefer many quick stimuli, preferably in the form of pictures. However, this is not only in social media, and these effects can be slowly seen with the next generation entering into adulthood. According to specialists from trade and marketing, Facebook will share the fate of six other enterprises. This potential scenario shows what challenges are posed to newly appearing online traders.

From the point of view of the enterprise, the Internet has become the basic platform for providing services and communicating with customers [9,10]. This does not apply only to e-shops. Now, the enterprises' websites or presence in social media have become one of the basic marketing elements and business contacts. Potential customers look for information and opinions about an enterprise on the web before they decide to buy its products or use its services.

E-commerce is an important driver of development, and it changes while running a business. It has transformed the market and reduced trade barriers for enterprises of all sizes, enabling sales at long distances and at different scales, thus leading to increased competition on the market [11,12]. Because of the rapid development in the world of e-commerce, enterprises in all parts of the world are able to provide basic services to the global community at affordable prices [13]. It can be said that practically all markets are open to the customer.

Many e-shops, in addition to the largest players on the market, operate as small and medium-sized enterprises. They are often created by small groups of friends, acquaintances, family, or individual persons, and over time they are expanded and modernized. E-commerce enables such enterprises to prepare a detailed description of their services, according to customers' expectations, along with the cost of such services and the expected time required to provide the order [14]. It is possible to quickly disseminate information about the offer and its changes.

E-commerce platforms also work for the benefit of customers who can easily withdraw from the purchase contract and can use various types of payments and shipments. From the point of view of the enterprise, the operating costs are lower, and the time needed to provide services is shorter. The convenience of online shopping and paying for services electronically is very valuable to all customers all over the world [15].

An important element that decides how well an e-shop functions is quality. In order to satisfy the needs of customers within e-commerce, enterprises that provide Internet services must remember to focus on quality in this highly competitive market. Therefore, knowledge about how to build

good relationships and maintain customer loyalty via the Internet has become an important issue for e-commerce [16]. Consequently, enterprises in this sector look for methods to improve quality.

All around us it is possible to notice issues related to the concept of sustainable development. Many enterprises in society try to operate and live according to the requirements of this concept, which has become very popular all over the world and in the sphere of the e-commerce economy [17]. Sustainable development has become popular among customers who look at the functioning of enterprises in this respect as well. Sustainability is, therefore, considered an effective way of maintaining competitiveness and attracting more consumers in virtual markets [18,19].

In the case of e-commerce, sustainable development models have specific features that influence particular dimensions (economic, social, and environmental) [20]. The use of sustainable development in e-commerce and the appropriate balance of individual dimensions may affect, for example, the efficiency and effectiveness of the enterprise (economic dimension), easier access to products or new jobs (social dimension), and less impact on the environment through less documents or waste (environmental dimension). Customers who increasingly follow the fashion for living in an eco-friendly way look for products and services from producers and suppliers who care about the natural environment and local communities.

People are more aware about environmental protection. It is not just a new fashion. Ecological awareness allows people to realize that every human being is responsible for the natural environment, its condition, degradation, and protection [21]. Many people pay attention to what they buy and what will happen to the products they purchase. They read labels more carefully to check the content of preservatives, artificial colors, or other substances included in the products. They check what the product packaging is made of and what will happen to this packaging after the product is unpacked [22]. So, environmental protection and sustainable development are elements that influence purchase preferences and customer behavior. That is why these issues in the case of e-commerce are so important.

The purpose of the paper is to identify, with use of a Kano questionnaire, the most important (according to potential customers) characteristics of a newly opened e-shop (located in the south of Poland) that sells organic products, which is to operate in a few Central European countries—Poland, Czech Republic, and Slovakia—with the possibility of expanding to other Central and Eastern European countries. The questionnaire included features related to sustainable development, and it was analyzed whether customers paid attention to them; that is, their environmental awareness was verified.

The survey created by the authors and presented in this paper is an original (authorial), universal study. It is a tool that allows the design, but also assessment, of the level of quality of the chosen e-shop and, thus, customer satisfaction in addition to their approach to sustainable development. It can be used or modified by other scientists to carry out similar research, and it is a contribution to the science of the quality of e-commerce services.

The rest of the article is organized as follows: Section 2 presents current practices (theoretical background) of e-commerce, service quality, and sustainable development. Section 3 includes the research problem and methodology. Sections 4 and 5 present the obtained results and their discussion. Finally, Section 6 presents the conclusions of the research.

## 2. Literature Review

### 2.1. E-Commerce

E-commerce is a big business that is getting bigger and bigger every day. It is a methodology of modern business that addresses the requirements of business organizations. It can be broadly defined as the process of buying or selling products with use of an electronic medium such as the Internet [23,24]. More and more enterprises are operating as a part of e-commerce, offering customers a whole range of products and services.

The e-service quality is "the degree to which an electronic service is able to effectively and efficiently fulfill relevant customer needs" [25]. E-services are, therefore, all the services that are offered on the Internet and from which the customer may choose. Customers more and more are using this form of shopping. E-services means not only convenience but, most of all, the ability to get things done without leaving home, practically at any time of the day.

What is more, over the years, the e-shops themselves, their offers, and their operations have changed, but also the shopping experience of these shops' customers has changed at the same time. Even more importantly, customers have become more and more demanding. While choosing a shop, they are guided by more and more different features of e-shops. It can be assumed that both e-shop and customer approaches will undergo further changes. As the authors of the E-commerce Report assume [26], soon the owners of online shops will face the challenges of not only how to gain new customers but, above all, how to build loyalty with those already present. The key feature will be not only the price but the convenience, timeliness, and quality of service.

On the other hand, it should be emphasized that the network economy represented by e-commerce has significantly contributed to economic development and has shown strong economic viability. More and more retailers and manufacturers use e-commerce platforms to sell their products directly to consumers, omitting subsequent intermediaries [27].

There are many factors that make e-commerce so popular. These include, for example, a wider selection of products/services, easy to search for products/services, attractive prices comparable to different suppliers, convenient shopping anytime and anywhere, affordable prices and often no delivery costs, easy return policies, higher customer satisfaction than in case of traditional shops, shortened delivery times, and lower delivery costs [28]. According to Anvari and Norouzi [29], the positive features of e-commerce are its low cost, efficiency, high sales, convenience, easy methods of personalizing information in accordance with consumer behavior, and the use of accessibility in social media.

Biener et al. [30] presented various socio-economic factors that are known to affect the development of e-commerce. They also drew attention to the fact that the skills necessary to effectively use information and communication technologies (ICT) are a key requirement for the effective adoption of e-commerce activities. Activities related to e-commerce require the availability of modern infrastructure, such as fiber optic technology [31], so that the transfer of datasets takes place in safe and fast environments [32].

E-commerce causes large economic growth, it allows the increase of profitability and number of customers, adds value to production, and, thus, leads to sustainable production activity. However, it should be mentioned that the costs of operation must include the costs of creating software and hardware, which is expensive in the initial phase (launching phase) [33]. On this basis, the following hypothesis was proposed.

**Hypothesis 1 (H1).** *The share of e-commerce in total trade will continue to increase due to many technical, market, or social factors and probably will soon exceed the share of stationary trade.*

*2.2. Service Quality in E-Commerce*

More and more people use e-shops. In Poland, there are 27.8 million Internet users, of which 56% have made at least one online purchase (54% in Polish e-shops, 23% in foreign ones). Most people shopping online are young (under 35 years). Among buyers, only 12% are people over 50 years old [34]. People appreciate this form of shopping, especially in the pre-Christmas period, preferably buying gifts at home and having a larger choice. As additional advantages, customers indicate speed (63%) and one-click payments (33%). Nearly half (49%) of e-consumers do shopping on working days between 5:00 p.m. and 9:00 p.m., which is when many traditional shops are already closed or are to be closed. Many people do it also during work hours [26]. According to EU forecasts, the development of a cross-border e-commerce model can bring the European economy EUR 415 billion a year, all while creating hundreds of thousands of new jobs [35].

Consumers from the Y generation (also called the Millennial generation, Millennials, Next Generation, iPod generation) are the largest global consumer group [36], the first generation to use advanced technologies [37], and are extremely aware of brands [38]. These are people who are characterized by widespread acceptance of the latest technologies, especially the Internet, mobile phones, and social media [39]. So, it is the generation that most often uses online shops. It can be assumed that the next generations will be even more open to new technologies and e-commerce [40]. As those from the Z generation grow up, some of them already entering adulthood, they will soon create their own families.

The biggest e-commerce challenge to understand the market is to chart the level of service quality according to customer perceptions [41]. The quality of electronic services is a basic determinant that influences the enterprise's long-term success [42].

An important feature of e-commerce is that customers during online shopping pay special attention to the sales platform and its operation [43]. Schmidt et al. emphasized that a well-designed website and excellent customer experience are the main factors behind the success of e-business luxury brands [44]. Kim claims that customers and the market require that Internet platforms constantly improve their services [45]. This is the first feature that determines the quality of e-commerce.

Consumers require faster and higher-quality services, which is why they require that the enterprises focus more on managing customer relationships. E-commerce services must be of high quality and reliable over time, which results in a positive delivery [17].

Regardless of the choice of features that decides the quality of the service, an important issue is the appropriate selection of a research method to assess quality and customer satisfaction. There are many methods described in the literature that can also be used by e-shops. They are, for example, Servqual and its variations, Servperf, importance/performance analysis, Critical Incidents Technique (CIT), mystery shopper, and Kano model, which are all used to assess the level of quality, but other methods should also be mentioned such as the customer satisfaction index, customer loyalty index, and net promoter score, which are used to assess the satisfaction and loyalty of customers.

These methods help to not only determine whether the customer is satisfied, but whether they will re-buy the product or re-buy the service. Thanks to their use, it is possible to indicate the strengths of the product or services and, above all, the reasons for customer dissatisfaction and, thus, the areas for potential improvement. Some of these methods, such as the Kano model, can be used in the design phase of a product or service before it is available on the market.

Measuring and monitoring quality assessments of e-services according to customers' opinions is considered critical because consumers' perceptions of quality affect the market share and profitability of e-sellers [46]. That is why it is worth it to be interested in quality from the very beginning of the enterprise's existence, the moment when the service is offered, and also during the design phase. On this basis, the following hypothesis was proposed.

**Hypothesis 2 (H2).** *Thanks to the appropriate selection of the research method to assess quality and customer satisfaction, it will be possible to determine the specific characteristics of a given service within e-commerce, which will positively affect customer satisfaction and the possible future of the enterprise.*

*2.3. Drivers of E-Commerce*

Personalization and customer experience are currently the most important elements that determine the success of e-shops. For several years, the e-commerce industry has been driven mostly by the development of mobile devices and changes in the way people use the Internet. However, since 2018, the number one trend has become personalization. This phenomenon, in practice, means the possibility of creating personalized offers and marketing messages.

Currently, customer experience (meaning the whole experience, but also user experience in contact with a given brand) plays an increasingly important role. In a huge number of e-shops, the customer looks not only for goods, but also for high-level service. Therefore, customer service is meant to make

them feel special, to keep up with their tastes and preferences with special attention to communication. Customer experience covers the entire purchasing process, from first contact with the offer, through the selection of the appropriate product variant, and to the rapid, safe payment for the goods and fast delivery. Preferences in relation to communication devices has also changed. The number of purchases by mobile devices has increased, while purchases from stationary devices (desktops) has decreased [47].

An important issue in the development of e-commerce services is machine learning and artificial intelligence. There are no studies yet on how machine learning and artificial intelligence can change the e-commerce industry, but there is no doubt that it will happen. Already today, leading trade enterprises use machine learning to process data and create personalized shopping paths for customers or in marketing [48]. Artificial intelligence helps in automation and simplifying sales. Advanced chat-bots increasingly facilitate communication between companies and consumers. Artificial intelligence can also be used to increase the cyber-security of online shops or in the recruitment process [49].

Marketing is a very important element that supports the efficiency of functioning of e-shops whether it is digital, social, content, or e-mail marketing. Legislative changes related to a significant limitation of trade on Sundays and holidays, in Poland for example [50], has caused a clear shift of marketing budgets from the offline to the online channel.

Large online shops have increasingly opted to open a traditional retail outlet, especially in big cities. We are dealing here with the phenomenon opposite to omni-channel, where online distribution channels are added to traditional ones. The boundary between traditional and online trade is obviously blurred [51,52]. Some examples of this trend may be the newly opened shops of online giants such as Casper or the clothing company Frank and Oak or how Amazon bought the Whole Foods supermarket chain. Market trends and requirements change so dynamically that the enterprises, in the fastest and the best ways, have adapted their offers to customer requirements so they can win. In such cases, the Kano model, which enables us to identify and analyze the attributes of customer satisfaction in an e-shop, may be very helpful. On this basis, the following hypothesis was proposed.

**Hypothesis 3 (H3).** *The e-commerce enterprises must learn about tools such as personalization, customer experience, machine learning, artificial intelligence, and omni-channels, and they must be able to use them properly in order to achieve success.*

*2.4. E-Commerce vs. Sustainable Development*

Where sustainable development is not ensured, there may be negative consequences that can lead to exhaustion of the ecosystem [53,54]. Therefore, e-commerce must also learn how to properly manage individual dimensions of sustainable development and adapt their own activities to it in order to achieve a balance between the two [20]. All three dimensions of sustainable development must, therefore, be linked to each other in order to bring both short- and long-term benefits [55]. Thanks to sustainable development in e-commerce, it is possible to increase operational efficiency and effectiveness, minimize resource use, reduce costs, benefit society by providing less harmful products and services in the best possible form, and create additional jobs [56].

There are some studies on sustainable development in e-commerce. Chaudhary [57] claims that it is useful for business organizations to look at the impact of e-commerce on the sustainability factors of the organization and see the benefits of this. From the point of view of the economy, the elements of sustainable development may be, for example, increased profitability and increased trade balance; for the society, for example, new jobs, new professions, gender equality, poverty reduction, and higher wages; and for the environment, for example, reduced depletion of substances that deplete the ozone layer, better air quality, reduced use of packaging materials, and so on [3]. Many negative effects can also be observed: from the point of view of the economy, high production or enterprise functioning costs, lack of efficiency, and low demand for products; for society, unemployment and problems with

general well-being; and for the environment, for example, negative impacts on biological communities, air pollution and water pollution in nearby areas, and so on [58].

Different authors [3] suggest that the obvious impact of e-commerce on sustainable development may include energy efficiency in cloud computing and use of IT, dematerialization (digitization) and collaboration techniques (*Customer Relationship Management* - CRM, chat, click to connect), travel limitations (videoconferencing, clicking on order), and improving the efficiency of the supply chain. What is more, the increasingly wider range of services provided by retailers causes transformation in the design of urban transport flows and the operation of vehicles in cities.

Tiwari and Singh claim that "due to vast and fast development of e-commerce, companies and businesses are paying so much attention to the production of low-cost products. They are not aware and, hence, not concerned about its adverse environmental implications... However, scientists and policy makers still do not have a clear statement about the relationship between e-commerce and the environment" [59].

Sustainable e-commerce has not yet been precisely defined. In papers [59,60], however, descriptions such as "conducting e-commerce that minimizes the environmental footprint of technological usage and promotes products or services that offer environmental and social benefits over traditional alternatives" can be found. This means that the goal of the website is crucial in determining its durability, and not just the technology implemented.

The development of e-commerce has led to an increase in orders delivered to customers, which results in high $CO_2$ emissions and even greater congestion in cities [61]. This situation will probably not change because of the high popularity of online shops. And it should be kept in mind that e-commerce operates according to specific laws. The customers may return the goods, due to the fact that they cannot physically check it before the purchase, which is also related to the transport of these products. A large amount of packaging is also needed for their transport. Goods sent to the customer must be properly packed and secured. The right packaging can help reduce greenhouse gas emissions and reduce the environmental impact [62].

There is an important, unfortunately negative, aspect related to the development of e-shops and their taking over the tasks of stationary shops. It concerns social issues. This is a form of digital exclusion for older people and people without access to the Internet. Many papers have touched on the issue of the greater digital exclusion of older people than younger ones [63,64]. A very big problem is that, in the majority of developed and developing countries, society is aging, which means high numbers of older, post-production people.

Digital exclusion involves unequal access and use of information and communication technologies (ICT), which are seen as necessary to fully participate in society [65]. It can, therefore, be concluded that older people, and those who do not have access to the Internet, do not have full access to products and services on the market. The offers addressed to them are limited to stationary shops and enterprises.

Some studies show how e-commerce can be less sustainable than its traditional counterpart in some spheres of activity [2–4]. Other studies also show that if e-commerce stays as it is, it will become unbalanced due to high energy consumption, data storage, and bandwidth and data center requirements [66].

Li et al. [67] claim that sustainable e-commerce can be achieved through better use of assets and control of production costs and factory administration. Thanks to its flexibility in responding in a timely manner to changing consumer requirements, e-commerce thereby increases customer satisfaction, loyalty, and trust through customer service support.

One important feature of e-commerce should be emphasized, namely, the fact that the customer, thanks to the Internet, can make purchases in e-shops from different countries. The main obstacle to conventional retailing is geographical distance. Geographical boundaries are less important on the Internet, but there are issues such as language, shipping costs, payment methods, and legal regulations [68,69], which can sometimes make shopping more difficult for Internet users.

Consumers are fighting for better health, ecological environment, and corporate social responsibility in their neighborhoods [70]. Therefore, an important element here is also corporate social responsibility (CSR), i.e., the concept that enterprises, at the stage of strategy building, take into account social interests and environmental protection as well as relations with various stakeholder groups [71]. Customers slowly force CSR attitudes from enterprises. And it should be remembered that CSR not only takes into account social responsibility but also the environmental and economic activities of the enterprise [72]. It should be underlined, then, how Hopkins claims [73] that CSR is a process to achieve sustainable development in societies.

It can, therefore, be concluded that in the case of e-commerce, sustainable development takes on a specific form. In each dimension of sustainable development, positive and negative factors can be distinguished (Figure 1). It will depend on enterprises from the e-commerce sector which factors will prevail and how they will affect particular dimensions of sustainable development.

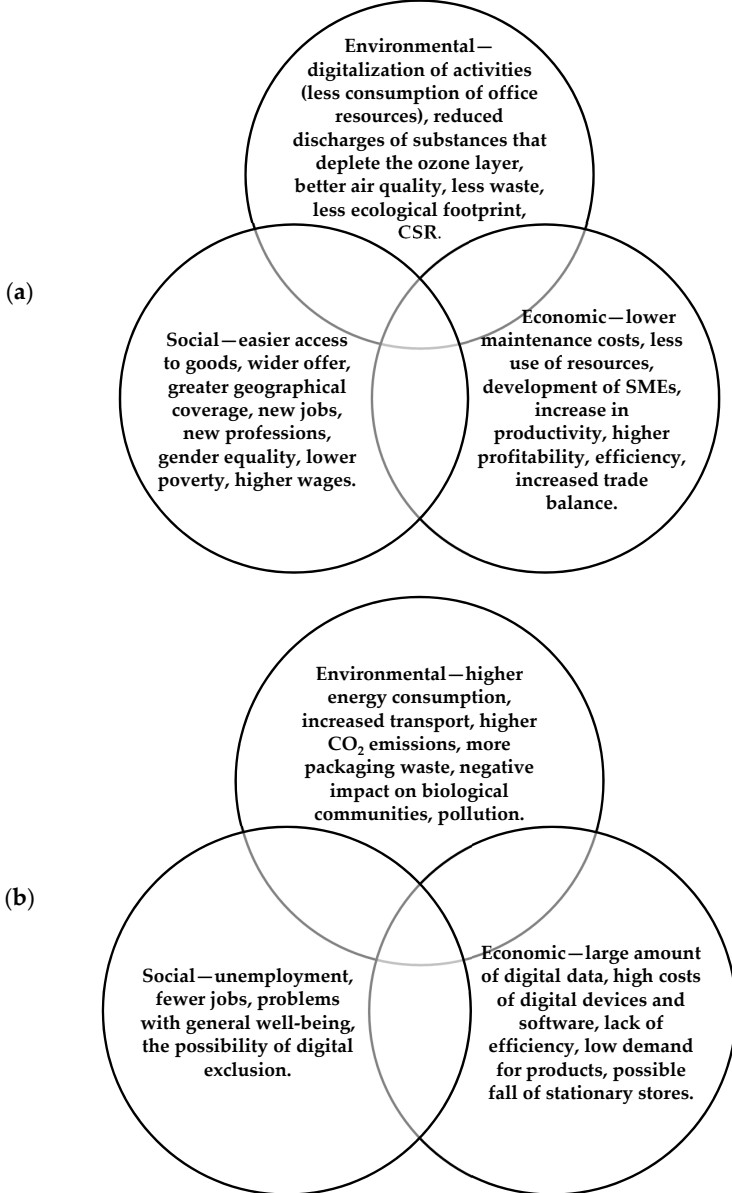

**Figure 1.** E-commerce vs. sustainable development: (**a**) advantages; (**b**) disadvantages [own study].

On this basis, the following hypothesis was proposed.

**Hypothesis 4 (H4).** *Each type of industry, including e-commerce, has specific factors in individual dimensions of sustainable development that can have a positive or negative impact on these dimensions and the enterprises themselves.*

## 3. Materials and Methods

### 3.1. Problems Description

How the online shop works, what features it is characterized by, and how these features are perceived by customers all affect the functioning of the shop and its financial result. That is why an important point in the e-shop's operation is the proper design of its so-called production system, including organization of activities and channels of communication with the customers.

The authors were asked to help create an e-shop (located in the south of Poland) with organic products, which is to operate in a few Central European countries: Poland, Czech Republic, and Slovakia, with the possibility of expanding to other Central and Eastern European countries. These are countries with similar cultures and similar markets, which is why the company decided to enter the market in the three above-mentioned countries.

Changing market conditions, large digitization, and Revolution 4.0 distinguish newly opened e-shops in the market, not only in terms of competitiveness, modernity, and customer-friendliness, but also in social and environmental terms. More and more customers consider whether enterprises care about the natural environment, and also the local community or operate in accordance with the concept of sustainable development.

Therefore, the authors decided that, with use of the assumptions of the Kano model, they would reach potential customers of the researched e-shop and define the basic features that this shop should be characterized with, taking into account selected assumptions of sustainable development.

### 3.2. Kano Model

The Kano model allows us to examine dependence on the development of a product or its service features and the level of customer satisfaction. It was developed in 1980 by Japanese professor Noriaki Kano [74]. With use of the Kano model it can be shown that not all components of a product are similarly important from the customers' point of view. Noriaki Kano categorized attributes into six groups [75,76]:

- "Must-be" or "must-have" attributes, which must be contained in the product and service as a standard. Lack of these features can lead to losing the customer. In the case of e-commerce, one of the "must-be" features is access to full contact information about the enterprise. It is not only good practice but also an element often checked by the customers. Nobody wants to buy goods from an anonymous source.

- "One-dimensional" attributes have the greatest importance to customer satisfaction from a product or service. It is critical that meeting this type of attribute is almost proportional to the level of customer satisfaction, which can translate into the frequency of purchasing the products or using the services. Problems with meeting "one-dimensional" attributes reduce customer satisfaction. However, this does not progress as fast as in the case of the "must-be" attributes. A "one-dimensional" feature can be, for example, the sorting options or the number of filters that the customer can choose. The more filters, the easier and faster the customer can make a choice.

- "Attractive" attributes have to attract the customer to a product or service. The popularity of these attributes is short-term and can be turned into the "must-be" attributes or disappear. Improperly designed "attractive" attributes can pass unnoticed. Their lack does not have an effect on customer satisfaction, but it may lead to the lack of his or her interest (if the customer is susceptible to the company continuously making the offer attractive). When it comes to e-commerce, there may be all types of promotions that will convince customers to buy in a given e-shop because of the lower price, which is much higher in other shops.

- "Reverse" attributes occur when the customer prefers absence of the attribute and does not like its presence. More of these attributes in the service or product leads to greater customer dissatisfaction. In the case of e-commerce, customers often do not want to pay for their orders before they arrive. They prefer to do it when the courier brings the package home.
- "Indifferent" attributes do not impact customer satisfaction. This may be, for example, the color of the website, which does not affect the functioning of website or content of the information. For the customer it is important to buy what they want, at the right price, and get the right information, and it is not important whether the site is green or blue.
- "Contradiction" occurs in the case of the Kano method. It is the attribute which was assessed as functional or non-functional in both forms of questions.

The Kano method was used in the form of a special type of questionnaire. The questionnaire contained positive and negative versions of statements/questions that referred to the attributes of a specific product [77]. Customers in the questionnaire indicated which features of the product or services must be included, which depended the level of customer satisfaction, and which characteristics customers did not want.

### 3.3. Research Description

Based on previous studies presented in many scientific papers, factors that were most often repeated were identified, i.e., factors that scientists, and probably customers, often refer to. Kim [45] collects information on the characteristics that determine the level of quality in e-commerce, analyzing the literature that is rich on the subject. Analyzing this paper, as well as other papers [17,43], one could indicate a group of factors that are most often described as those that most affect customer satisfaction in e-commerce, among which are the website, its operation and content, customer relationships, reliability, attractive prices, a wide range of products, and returns and payment methods. Among e-commerce factors that are related to sustainable development, and which are presented in [20,54–64], one should mention, among others, packaging waste, transport, CSR, lower poverty, access to goods, wide offers, fixed defects, and problems with general well-being. These factors helped to create a survey on which the conducted research was based.

The first stage of research was to create a survey form. This questionnaire consisted of two parts: characteristics of respondents and a set of attributes for the research e-shop. The characteristics of the respondents included features such as: sex, age, country, employment status, place of residence, and monthly income per person. It allowed us to characterize people who took part in the research. In the second part of the survey, potential attributes, which should characterize the research shop, were included. Some of the attributes were related to the functioning of the e-shop in accordance with the principles of sustainable development. The respondents were asked to evaluate the attributes in cases when they will occur (positive attributes) and when they will not occur (negative attributes) in the research e-shop.

From the answers, it was possible to indicate the features that must be included (must be) but also those features that affect overall customer satisfaction (one-dimensional). A list of the positive attributes of the Kano questionnaire is presented in Table 1.

Attributes were chosen in such a way that they could be used in the future for broader research and design of other e-shops, also of different types, and not necessarily an e-shop with organic products. Evaluations of these attributes (answers to these questions) were based on the following scale: (a) "I like it", (b) "That is the way it has to be", (c) "I do not mind it", (d) "I can put up with it", (e) "I do not like it".

**Table 1.** List of attributes for the research e-shop [own study].

| Attribute Number | Attributes (Positive Attributes) |
|---|---|
| 1 | New e-shop should have a well-organized and clear website. |
| 2 | New e-shop should have entertaining pages in the main social media. |
| 3 | Information included on the website of the new e-shop should be up to date and comprehensive. |
| 4 | The website of the new e-shop should include the exact contact details. |
| 5 | Offers of the new e-shop should be updated often. |
| 6 | Categories in the offers of the new e-shop should be clear and logical. |
| 7 | Information about the products of the new e-shop should be complete and their images legible. |
| 8 | Information included on the website of the new e-shop should be accurate. |
| 9 | Regulations of the new e-shop should be accessible and understandable. |
| 10 | The website of the new e-shop should include a detailed description of the purchase process. |
| 11 | The website of the new e-shop should include an easy-to-use search engine. |
| 12 | The website of the new e-shop should include options to filter results. |
| 13 | The website of the new e-shop should include purchase history options. |
| 14 | Ordering options in the new e-shop should be clear and easy to use. |
| 15 | Registration in the new e-shop should be required to make purchases. |
| 16 | All customer data provided on the website of the new e-shop during the order should be secure and remain private. |
| 17 | A customer in the new e-shop should be able to choose different payment methods. |
| 18 | A customer in the new e-shop should be able to choose different forms of delivery. |
| 19 | A customer in the new e-shop should be able to easily contact support to obtain additional information about the products. |
| 20 | A customer in the new e-shop should be able to easily contact support for the order to explain any inconsistencies. |
| 21 | A customer in the new e-shop should be able to cancel the transaction before its implementation. |
| 22 | Deliveries in the new e-shop should be carried out in accordance with the terms indicated on its website. |
| 23 | A customer in the new e-shop should be able to track their purchase. |
| 24 | Supply of goods from the new e-shop should be realized during hours that are convenient for the customer. |
| 25 | A customer of the new e-shop should be able to refuse to accept the consignment in case of any inconsistencies. |
| 26 | Products that are sent to customers by the new e-shop are safely packed. |
| 27 | The packaging of products sent by the new e-shop is easy to re-use, e.g., to send back the product. |
| 28 | The packaging of products sent by the new e-shop contains relevant documents (invoice or bill and a document allowing the customer to return the goods). |
| 29 | Financial transactions related to the purchase in the new e-shop should be easily made. |
| 30 | After purchase in the new e-shop, a customer should be able to add their opinions about the purchase process. |
| 31 | After purchase in the new e-shop, a customer should be able to add their opinions about carrier and delivery. |
| 32 | A customer in the new e-shop should be able to return the product if they did not like it. |
| 33 | A customer in the new e-shop should be able to return or exchange products that were damaged during transport. |
| 34 | A customer in the new e-shop should be able to have a product repaired under warranty. |
| 35 | The new e-shop should introduce loyalty programs. |
| 36 | The new e-shop should have a friendly employment policy. |
| 37 | Employees of the new e-shop work in decent conditions for a decent wage. |
| 38 | The new e-shop should participate in regional ecological actions. |

The surveys were made available by various social media and professional contacts of the authors, and it was performed from September 2018 to March 2019. A total of 1101 responses were obtained (Czech Republic 314, Poland 521, and Slovakia 266). In total, 32 questionnaires were rejected because they were not filled completely (some answers were missing, so comparison of answers was not possible). It should be emphasized that the survey was long (38 attributes), and respondents were reluctant to devote their valuable time to such long surveys. It should also be noted that the survey concerned an e-shop with organic products, i.e., specific products targeted at a specific customer group. There is a fashion among young people to be fit and eco.

Analysis of the results was based on the individual types of attributes contained in the questionnaire using the comparisons presented in Table 2. Next, it was checked which type of feature was indicated the most often.

**Table 2.** Types of attributes in the Kano method [78].

| | | Negative | | | | |
|---|---|---|---|---|---|---|
| | | I like it | That is the way it has to be | I do not mind | I can put up with it | I do not like it |
| **Positive** | I like it | Q | A | A | A | O |
| | That is the way it has to be | R | I | I | I | M |
| | I do not mind | R | I | I | I | M |
| | I can put up with it | R | I | I | I | M |
| | I do not like it | R | R | R | R | Q |

Notes: A—attractive; O—one-dimensional; M—must-have; I—customer was indifferent to the attribute; R—customer did not like the attribute; Q—there was a contradiction: customers both wanted the attribute to occur and not to occur.

The assessment given by the customers in the Kano questionnaire could be used to calculate customer satisfaction and dissatisfaction indexes. These indexes are given by [79] the following equations:

$$\text{satisfaction index} = \frac{A+O}{A+O+M+I}, \tag{1}$$

$$\text{dissatisfaction index} = -\frac{O+M}{(A+O+M+I)}. \tag{2}$$

A minus sign was added to the dissatisfaction indexes (Equation (2)) in order to emphasize the negative effects on customer perception if the quality of the product was poor. The satisfaction index was in the range of (0, 1). If the value was close to 1, customer satisfaction was very high. If the value was close to −1, customer dissatisfaction was very high [80].

Interpretation of indexes can be done in a graphical manner. For this purpose, a two-dimensional matrix was created, where the X-axis was the index of dissatisfaction for individual attributes in absolute terms, and the Y-axis was the satisfaction index. The discussion of results was made on the basis of Table 3.

**Table 3.** Interpretation method [79].

| Distribution of Response | XY Pair | Location of the Point on the Graph |
|---|---|---|
| All attractive | 0, 1 | Top left corner |
| All one-dimensional | 1, 1 | Top right corner |
| Evenly split between attractive and one-dimensional | 0.5, 1 | Middle of the top, halfway between attractive and one-dimensional—point A |
| All must-have | 1, 0 | Bottom right corner |
| Evenly split between one-dimensional and must-have | 1, 0.5 | Middle of right edge, halfway between one-dimensional and must-have—point B |
| All indifferent | 0, 0 | Bottom left corner |
| Evenly split between must-have and indifferent | 0.5, 0 | Middle of bottom edge, halfway between must-have and indifferent—point C |
| Evenly split between indifferent and attractive | 0, 0.5 | Middle of left edge, halfway between indifferent and attractive—point D |
| Evenly split among attractive, one-dimensional, must-have, and indifferent | 0.5, 0.5 | Exact middle of graph—point E |
| Evenly split between attractive and must-have | 0.5, 0.5 | Exact middle of graph, halfway between attractive and must-have, without an influence of one-dimensional or indifferent—point E |
| Evenly split among attractive, one-dimensional, and must-have | 0.67, 0.67 | Equally spaced between attractive and must-have, but influenced by one-dimensional—point F |

## 4. Results

### 4.1. Characteristics of Respondents

A total of 1101 respondents took part in the survey. A total of 1069 questionnaires were considered for further analysis. Thirty-two questionnaires were not filled in completely (comparison of answers was not possible), so the contained information was not fully reliable; therefore, these answers were omitted.

The analysis of characteristics of the respondents participating in the research (Table 4) showed that the survey was more often completed by women (57% of responses). Important information about potential customers was obtained in the case of analyses of the age of respondents. Older people constituted a small percentage of respondents (9% of people aged 50–60 and 4% of people aged over 60). These people are less likely to use the Internet and social media. It can also be assumed that this was an element of digital exclusion. The respondents were mostly young people, up to 40 years old. Most often the surveys were filled in by full-time workers (36%) and students (23%). Respondents were mainly residents of a city with 500,000 inhabitants or more (31%) or a city with 200,000–500,000 inhabitants (27%). Monthly income per person was 300–800 Euro (39%) or 800–1500 Euro (32%), that is, people with average material status. It should be reminded again that the study covered an e-shop with organic products. These are products that are popular, especially among young people, for those who want to live healthy lives.

**Table 4.** Characteristics of respondents [own study].

| Category | Possible Answer | Results (%) |
|---|---|---|
| Sex | Female | 57 |
| | Male | 43 |
| Age | Up to 20 years old | 17 |
| | 20–30 years old | 29 |
| | 30–40 years old | 27 |
| | 40–50 years old | 14 |
| | 50–60 years old | 9 |
| | Over 60 years old | 4 |
| Country | Czech Republic | 28 |
| | Poland | 48 |
| | Slovakia | 24 |
| Employment Status | Student | 23 |
| | Full-time worker | 36 |
| | Part-time worker | 11 |
| | Trainee | 3 |
| | Disability pensioner/retired | 4 |
| | Unemployed | 7 |
| | Self-employed | 16 |
| Place of Residence | City 500,000 inhabitants or more | 31 |
| | City 200,000–500,000 inhabitants | 27 |
| | City 100,000–200,000 inhabitants | 19 |
| | City less than 100,000 inhabitants | 17 |
| | Countryside | 6 |
| Monthly Income per Person | 300 Euros or less | 11 |
| | 300–800 Euros | 39 |
| | 800–1500 Euros | 32 |
| | 1500–2200 Euros | 15 |
| | 2200 Euros or more | 3 |

*4.2. Kano Results*

Answers of individual respondents obtained in the surveys were compared in pairs (positive and negative attributes) in accordance with the assumptions presented in Table 3. The type of feature that occurred most frequently was indicated, and satisfaction and dissatisfaction indexes for individual attributes were calculated. The attribute numbers corresponded to the numbers and attribute names from Table 1. A comparison of the results obtained according to the Kano model is presented in Table 5. As in Figure 2, a list of different types of attributes according to the assessment of the attribute was also presented.

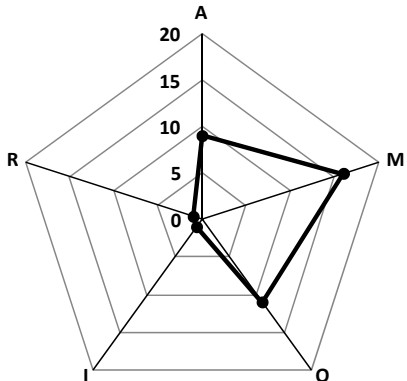

**Figure 2.** List of attribute types, where: A—attractive; O—one-dimensional; M—must-have; I—indifferent; and R—reverse [own study].

**Table 5.** Kano questionnaire results [own study].

| Attribute Number | Number of Answers in the Category | | | | | | Assessment of the Attribute | Satisfaction Index | Dissatisfaction Index |
|---|---|---|---|---|---|---|---|---|---|
| | A | M | O | I | Q | R | | | |
| 1 | 128 | 826 | 97 | 16 | 2 | 0 | M | 0.21 | −0.87 |
| 2 | 694 | 153 | 174 | 44 | 1 | 3 | A | 0.82 | −0.31 |
| 3 | 146 | 793 | 117 | 13 | 0 | 0 | M | 0.25 | −0.85 |
| 4 | 187 | 594 | 267 | 21 | 0 | 0 | M | 0.42 | −0.81 |
| 5 | 226 | 324 | 472 | 46 | 1 | 0 | O | 0.65 | −0.75 |
| 6 | 196 | 217 | 614 | 39 | 3 | 0 | O | 0.76 | −0.78 |
| 7 | 518 | 349 | 187 | 14 | 0 | 1 | A | 0.66 | −0.50 |
| 8 | 539 | 337 | 186 | 7 | 0 | 0 | A | 0.68 | −0.49 |
| 9 | 163 | 698 | 203 | 5 | 0 | 0 | M | 0.34 | −0.84 |
| 10 | 117 | 456 | 493 | 3 | 0 | 0 | O | 0.57 | −0.89 |
| 11 | 211 | 285 | 203 | 369 | 1 | 0 | I | 0.39 | −0.46 |
| 12 | 560 | 106 | 317 | 86 | 0 | 0 | A | 0.82 | −0.40 |
| 13 | 462 | 364 | 202 | 41 | 0 | 0 | A | 0.62 | −0.53 |
| 14 | 317 | 94 | 634 | 24 | 0 | 0 | O | 0.89 | −0.68 |
| 15 | 17 | 71 | 116 | 83 | 3 | 779 | R | 0.46 | −0.65 |
| 16 | 38 | 952 | 79 | 0 | 0 | 0 | M | 0.11 | −0.96 |
| 17 | 59 | 752 | 247 | 11 | 0 | 0 | M | 0.29 | −0.93 |
| 18 | 176 | 654 | 226 | 9 | 1 | 3 | M | 0.38 | −0.83 |
| 19 | 374 | 274 | 403 | 18 | 0 | 0 | O | 0.73 | −0.63 |
| 20 | 269 | 349 | 397 | 54 | 0 | 0 | O | 0.62 | −0.70 |
| 21 | 269 | 243 | 361 | 167 | 0 | 29 | O | 0.61 | −0.58 |
| 22 | 415 | 399 | 184 | 71 | 0 | 0 | A | 0.56 | −0.55 |
| 23 | 348 | 264 | 199 | 258 | 0 | 0 | A | 0.51 | −0.43 |
| 24 | 52 | 674 | 339 | 3 | 1 | 0 | M | 0.37 | −0.95 |
| 25 | 23 | 789 | 204 | 53 | 0 | 0 | M | 0.21 | −0.93 |
| 26 | 217 | 376 | 426 | 49 | 0 | 1 | O | 0.60 | −0.75 |
| 27 | 146 | 194 | 397 | 332 | 0 | 0 | O | 0.51 | −0.55 |
| 28 | 103 | 552 | 391 | 16 | 0 | 7 | M | 0.47 | −0.89 |
| 29 | 159 | 423 | 473 | 14 | 0 | 0 | O | 0.59 | −0.84 |
| 30 | 397 | 337 | 95 | 301 | 1 | 38 | A | 0.44 | −0.38 |
| 31 | 369 | 164 | 114 | 265 | 0 | 157 | A | 0.53 | −0.30 |
| 32 | 33 | 957 | 53 | 26 | 0 | 0 | M | 0.08 | −0.94 |
| 33 | 23 | 983 | 47 | 14 | 2 | 0 | M | 0.07 | −0.97 |
| 34 | 34 | 994 | 34 | 7 | 0 | 0 | M | 0.06 | −0.96 |
| 35 | 89 | 157 | 676 | 147 | 0 | 0 | O | 0.72 | −0.78 |
| 36 | 97 | 653 | 119 | 198 | 2 | 0 | M | 0.20 | −0.72 |
| 37 | 39 | 511 | 277 | 241 | 0 | 1 | M | 0.30 | −0.74 |
| 38 | 93 | 684 | 136 | 156 | 0 | 0 | M | 0.21 | −0.77 |

Where A—attractive; O—one-dimensional; M—must-have; I—indifferent; and R—reverse.

When analyzing the results presented in Table 5, it can be noticed that the respondents' answers were very diverse. Political changes that have taken place in Central and Eastern Europe in the last 30 years, as well as changes in the market, have also caused changes in customer behavior. There is a lot of competition on the market, and customers know that they can choose, that it is up to them to decide whether to buy a particular product or use a particular service. That is why customers have become very fussy and demanding, which was reflected in the results.

Most of the attributes of the research e-shop are, according to customers, must-have features (16 out of 38 attributes). These are attributes that have to be included in the final list of the attributes of the research e-shop. Their presence will determine whether customers opt for new e-shop services. In Table 5 the must-have features are marked with the letter M. Lack of such features means that customers go somewhere else.

In analyzing the research results, including "must-be" features, it could be concluded that, in the case of the research e-shop, respondents indicated features frequently repeated in previously published papers about quality in e-commerce, i.e., well-organized, up-to-date and comprehensive website, secure and private staff to handle customer data, possibility to choose the way of delivery, possibility to return the product, and certainty of repairing a broken product.

Another 11 attributes were marked as one-dimensional features (marked with the letter O), i.e., those that determine the level of customer satisfaction. When this feature is a better fulfilled, the customer is more satisfied. These are not obligatory features, i.e., their absence should not affect the decision to buy goods from another e-shop. But it is important to determine the appropriate level of these attributes to be sure that customers are satisfied enough.

Nine attributes, according to the customers of the research e-shop, were the attractive features (designation A). Their presence may influence customer decisions to use the offer of the research e-shop or another competitive one. These features can distinguish the research e-shop from the crowd of others on the market. Therefore, it is important that these features are included in the design of the new e-shop because they will attract customers. What is more, in a short time, when customers get used to the high level of these features, these attributes can become must-have features.

Interestingly, in the case of several attributes, singular contradictions in the responses of some respondents were noted. This means that they marked the answers "I like it" or "I do not like it" in the positive and negative version of the attributes (see Table 2). This was probably a mistake of respondents who quickly wanted to complete the questionnaire, or respondents did not fully read the attribute at the time of its assessment. Fortunately, as mentioned before, it did not happen too often.

Attribute 15, e.g., "Registration in the new e-shop should be required to make purchases" was a reverse feature. Most respondents do not want to register to take advantage of the new e-shop offer. The purchases without registration are faster for the customer, and, at the same time, they do not have to provide too much private information. It also allows a greater freedom of shopping.

Attribute 11, e.g., "The website of the new e-shop should include an easy-to-use search engine" by most respondents was marked as an indifferent feature. This means that it does not affect the satisfaction or dissatisfaction of customers. They do not pay too much attention to this attribute, so when designing a new e-shop, it can be treated less seriously.

As mentioned earlier, a few questions regarding the e-shop in the Kano questionnaire in terms of sustainability were added. These were the attributes of two dimensions of sustainable development: environmental and social.

As for the environment, these were attributes 18, 26–28, 32–34, and 38. Attributes 26–28 were associated with packing the ordered goods. Customers want to be sure that the packaging of the goods is "ecological" and can be reused. Attributes 18 and 32–34 were related to the shipment of the ordered goods to the customer or its return to the e-shop. The customer wanted to be able to consciously choose a greener means of transport. Attribute 38, e.g., "New e-shop should participate in regional ecological actions" was associated with pro-ecological activities and the impact of e-shop activities on its stakeholders. Attributes 26–27 were one-dimensional features, while the remaining attributes were must-have features, i.e., environmental issues were important according to respondents and must be included in the research e-shop.

In the case of the social dimension, the attributes were related to three issues: different payment methods for the goods (attributes 17 and 29), data security (16), and the working and pay conditions of employees (36–37). Attribute 29 was a one-dimensional feature, and the remaining attributes were must-have features.

It can be concluded that respondents understand the need for sustainable development, and the features associated with it were treated by them mostly as must-have features, which means they must be included among the attributes of the new e-shop.

Comparing the results of this research with previous studies available in the literature, it could be seen that respondents pointed to certain features of the research e-shop, in terms of sustainable development, that were repeated in other studies: safe, reusable packaging (waste), possibility to return the product and to have it fixed in case of any problems (transport), regional ecological actions (CSR), different methods of payment (lower poverty), and friendly employment policies (problems with general well-being).

In the case of satisfaction index, the maximum value was 0.89 (attribute 14); it was a one-dimensional feature. In addition, in two cases (attributes 2 and 12) the value of this index was above 0.8, and the attributes were attractive features.

In many cases, the dissatisfaction index was close to −1, and in the case of seven attributes this index was in the range (−1, −0.9). These were must-have features with a very large predominance of indications for this type of attribute.

## 5. Discussion

Satisfaction and dissatisfaction indexes for individual attributes allowed us to create a map of attributes and a more precise indication of the type of attributes. This map helped in the indication of must-have attributes and other types of attributes. The map of attributes for this research enterprise is shown in Figure 3.

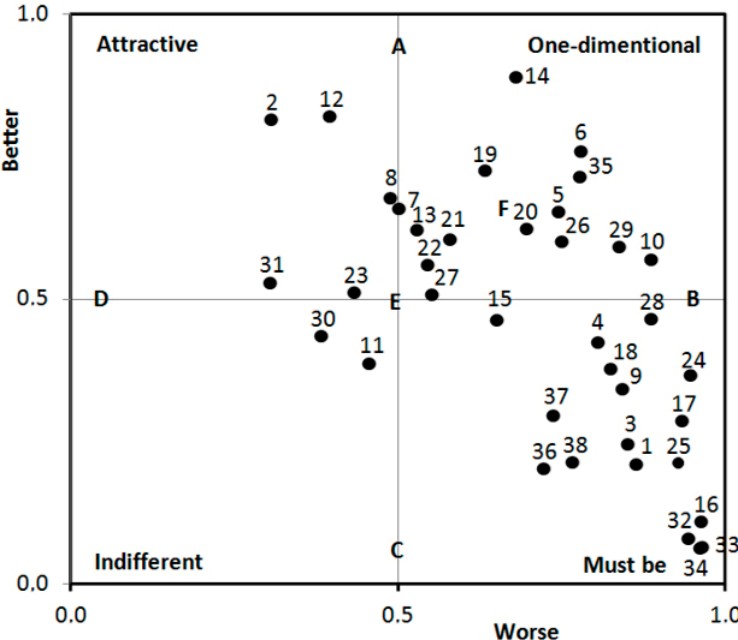

**Figure 3.** Map of attributes according to the Kano questionnaire [own study].

According to the map shown in Figure 3, it can be seen that many attributes were, in reality, a mix of features. Most of the points were placed on the right side of the map (high dissatisfaction index), with the majority of points in the middle of that half of the map (i.e., middle of the right edge, halfway between one-dimensional and must-have, according to Table 3). Many points were located near the "must-be" point. The least number of attributes was in the lower left square. None of the points were placed in the lower left corner, called "indifferent".

Regarding attributes 16, 32, 33, and 34, which were located in the lower right corner of the map, it can be said that they were purely must-have features, as already indicated in Table 5. These attributes must absolutely be among the attributes of the new e-shop, otherwise customers will not want to use its services.

Among the other features, it was impossible to find one that purely (a very large proportion of the answers) predominated over the other types of features contained in the map. The three remaining other corners of the map were empty. Many points were placed in the middle of the map. This means the responses were very diverse among potential customers, so it was not entirely possible to say what type of attribute they prefer. This is the most difficult problem to solve because, in this case, it is difficult to satisfy all customers.

## 6. Conclusions

Turbulent environments and high competition on the market force enterprises to introduce changes and adapt to constantly changing customer requirements. One of the strongest factors that forces huge changes in enterprises is Revolution 4.0. Among the tools of Industry 4.0, particular attention should be paid to the Internet. Nowadays, enterprises that do not operate on the Internet are less visible because most customers look for information and goods on it.

Another factor that forces change in enterprises is the increasingly common concept of sustainable development. These changes are aimed at more rational and efficient management of all resources that will allow for less pressure and less impact on the environment. Sustainable development is derived from the triple bottom line concept, which assumes a balance between the three pillars (dimensions): an environment that is uncontaminated and exploited in an appropriate manner, but necessary to run a business and maintain people's quality of life; a society that seeks to ensure human rights and equality, preserve cultural identity, and respect cultural race and religion diversity; and economic durability necessary to maintain the natural, social, and human capital required for income and standards of living. Sustainable development is a particular challenge for small and medium-sized enterprises [81,82].

In the case of e-commerce, particular dimensions of sustainable development have specific positive and negative elements. All three dimensions of sustainable development must have links with each other in order to bring both short- and long-term benefits, and that the enterprise (in this case an e-shop) gets a balance between all these dimensions. This will increase operational efficiency and effectiveness, minimize resource use and reduce costs, provide less harmful products and services in the best possible form, reduce the impact on the natural environment, and create additional jobs, especially in the case of small and medium-sized enterprises.

In this paper, a Kano questionnaire was used to indicate the basic features that the newly opened e-shop with organic products, which is supposed to operate in several countries of Central Europe, should have, taking into account the selected assumptions of sustainable development.

Thanks to the surveys, in which potential customers from the Czech Republic, Poland, and Slovakia took part, 16 must-have features, which must be included among the features of the research e-shop, and 11 one-dimensional features have been identified that will affect the satisfaction of potential customers. The respondents also indicated a reverse feature, i.e., one that they do not want as an e-shop feature (that registration in the new e-shop should not be required to make purchases), and one indifferent feature (the website of the new e-shop should have an easy-to-use search engine), which do not affect the satisfaction or dissatisfaction of potential customers. The features associated with sustainable development were mostly noticed by customers and marked as must-have features. It should be emphasized, therefore, that the respondents have an ecological awareness and see the need that the research e-shop operates in accordance with the concept of sustainable development.

The research allowed, therefore, to indicate the features that must be taken into account for the customers to use the services of the research e-shop and also the features influencing customer satisfaction. The success of the research e-shop will depend on them. Only satisfied customers will want to use the services of the research e-shop more often. What is more, it should be remembered that only a satisfied customer can become a permanent customer, which means their loyalty to the e-shop. In addition, a satisfied and loyal customer means a good source of advertisement for the enterprise and the possibility to attract other customers, which should be reflected in the achieved profits and, thus, affect the success of the e-shop.

These results were given to the management of the research e-shop and were included during the design of its operation. The management agreed that after 1–2 years of operation of the e-shop, the authors will be able to carry out new research to assess the functioning of the indicated features and check the operation of the e-shop in conditions of sustainable development.

The methodology used in the paper is very universal. The Kano questionnaire created for the needs of this research can be used by other e-shops, also of different types, for the design of their activities and the quality assessment of the offered services.

Of course, the research was not without limitations. The survey was created on the basis of information obtained from the managers of the designed e-shop and the authors' experience. Some important factors could have been omitted due to the subjectivity of authors and managers. The survey, as mentioned earlier, was made available by various social media and professional contacts of the authors. This could have influenced the group of respondents. The sample size could also be criticized as too small and the research period too short, which was due to restrictions on the part of managers. This is why revised results after 1–2 years of operation of the e-shop are needed, with possibility to change some elements of the survey if needed. Another limitation was the fact that customer requirements, and the market itself, are constantly changing, so results after a short time can be outdated. Therefore, the revision of the results should be repeated every once in a while.

**Author Contributions:** Both authors participated equally in the preparation of this article. R.U. created the main idea of the research; M.I. created the methodology. Both authors conducted the experiment and developed the results.

**Funding:** This research received no external funding.

**Conflicts of Interest:** The authors declare no conflict of interest regarding the publication of this article.

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
