# Peer review of "How to Make E-Commerce More Successful by Use of Kano’s Model to Assess Customer Satisfaction in Terms of Sustainable Development"

_sustainability, doi:10.3390/su11184830_

Round 1
Reviewer 1 Report
My comments on the paper - How to make e-commerce more successful by use of Kano's model to assess customer satisfaction in terms of sustainable development. Case study- are as follows.
- The goal of the paper and main results are included in the abstract.
- Keywords are appropriate.
- The structure of the paper is appropriate.
- The introduction provides the necessary background information, but it must be stated the added value that the paper brings to the existing academic literature.
- In the introduction section, the structure of the paper on sections is not provided.
- The research methodology used by the author is adequate for the approached subject
- The results of the research are clearly underlined.
- It is not pointed out if the results of the research are in accordance or not with other studies. We recommend that the results obtained from the study should be compared with the results obtained in the case of similar researches from the academic literature.
- In section 3, the author does not indicate the time period for which the questionnaire is analyzed.
- The conclusions are significant, but the author does not mention the limits of the research.We consider that the author can show the limitations of the analysis carried out in his paper.
- The references used by the author are appropriate.
- References: follow the journal guidelines.
Reviewer 2 Report
I would like to thank the Editor for the invitation to review this paper.
I have read the paper with interest. I found the paper is interesting and written using academic standards.
Some suggestions to improve it before publishing
Abstract: I suggest to cut off the study questions and develop the presentation of study results. Give some more info on study sample. Is it really case study if it is written “the Respondents indicated 16 must-be …” it is rather survey Introduction is quite proper and informative Literature review seems to me too long using too may pointed not discussed sources (I am sure many of them are not necessary, I suggest to leave 50%=60% of the most important literature. Also this part should be reconsidered to provide some hypothesis based on the literature review. Methodology and results are quite interesting and properly presented There is lack of study results discussion in relation to previously published results There are no study limitations and future study discussion English must be professionally correctedAuthor Response
Please see the attachment.

Round 2
Reviewer 2 Report
I am quite satisfied of the improvements.
Congratulation to the authors.
Great job!